# Australian Smokers’ Sensory Experiences and Beliefs Associated with Menthol and Non-Menthol Cigarettes

**DOI:** 10.3390/ijerph18115501

**Published:** 2021-05-21

**Authors:** Serafino Mancuso, Emily Brennan, Kimberley Dunstone, Amanda Vittiglia, Sarah Durkin, James F. Thrasher, Janet Hoek, Melanie Wakefield

**Affiliations:** 1Centre for Behavioural Research in Cancer, Cancer Council Victoria, Melbourne, VIC 3004, Australia; sam.mancuso@cancervic.org.au (S.M.); kimberley.dunstone@cancervic.org.au (K.D.); amanda.vittiglia@cancervic.org.au (A.V.); sarah.durkin@cancervic.org.au (S.D.); melanie.wakefield@cancervic.org.au (M.W.); 2Department of Health Promotion, Education, and Behavior, Arnold School of Public Health, University of South Carolina, Columbia, SC 29208, USA; THRASHER@mailbox.sc.edu; 3Department of Public Health, University of Otago, Wellington 6140, New Zealand; janet.hoek@otago.ac.nz

**Keywords:** menthol, menthol ban, tobacco products, sensory experiences, harm perceptions, public policy

## Abstract

Many current smokers incorrectly believe that menthol cigarettes are less harmful, likely due to the biological and sensory effects of menthol, which can lead smokers to have favourable sensory experiences. In this study, we measured the extent to which Australian smokers associate certain sensory experiences with smoking menthol and non-menthol cigarettes, and their beliefs about how damaging and enjoyable they find cigarettes with each of these sensory experiences. A sample of 999 Australian 18–69-year-old weekly smokers was recruited from a non-probability online panel; this study focuses on the 245 respondents who currently smoked menthol cigarettes at least once per week. Current menthol smokers were four to nine times more likely to experience menthol rather than non-menthol cigarettes as having favourable sensory experiences, including feeling smooth, being soothing on the throat, fresh-tasting and clean-feeling. Menthol smokers perceived cigarettes with these favourable sensations as less damaging and more enjoyable than cigarettes with the opposite more aversive sensory experience. Efforts to correct these misperceptions about risk will likely require messages that provide new information to help smokers understand that these sensations do not indicate a lower level of risk. Banning menthol in tobacco products—as has recently been done in some nations—would also be a timely and justified strategy for protecting consumers.

## 1. Introduction

Menthol has been used as a characterising flavour additive in tobacco products since the late 1920s [1,2,3]. Menthol in cigarettes is associated with increased initiation and progression to regular cigarette smoking [1,2,3]. For example, young people who initiate smoking with menthol cigarettes are at greater risk of progressing to regular smoking and nicotine dependence than young people who initiate with non-menthol cigarettes [4]. Likewise, menthol cigarette use has been found to nearly double the odds of progression from nondaily to daily smoking amongst young adults aged 18 to 34 years [5]. Menthol also appears to make it more difficult for some established smokers to quit smoking [6,7] and may increase the addictiveness of nicotine by conditioning the craving for nicotine and enhancing the reinforcing actions of nicotine in the brain [1,2]. A recent systematic review reported that 9 out of 14 studies found greater nicotine dependence among menthol smokers compared to non-menthol smokers [8]. This increased nicotine dependence may contribute to lower smoking cessation rates among menthol smokers relative to non-menthol smokers, even though menthol smokers have relatively stronger quit intentions and have made a greater number of quit attempts [8].

Menthol cigarettes have historically been marketed as safer and healthier to smoke than non-menthol cigarettes [3]. The effects of this historical marketing persist, as many menthol smokers continue to incorrectly believe that menthol cigarettes are safer or less harmful [3,9,10,11,12]. For example, Keller et al. [13] found that current menthol smokers were more likely to believe that menthol cigarettes were better for a sore throat, healthier, contained fewer chemicals and were less harmful than non-menthol cigarettes.

In addition to the effects of marketing, these misperceptions may also arise from the impact that taste and other sensory experiences have on smokers’ perceptions of risk [14]. Since the menthol in tobacco smoke has cooling, anaesthetic and analgesic properties, it can make menthol cigarettes easier and more palatable to smoke [2,6,15,16,17]. For example, compared to non-menthol cigarettes, smokers tend to experience menthol cigarettes as smoother, more soothing on the throat, fresher tasting and having a cleaner taste [9,11,12]. Importantly, smokers may interpret these positive sensory experiences as indicating reduced risk, which may increase the likelihood they will smoke menthol cigarettes [1,2]. For example, 16% of Malaysian and 35% of Thai smokers believed that menthol cigarettes were less harmful due to their perceived smoother smoke [18]. Likewise, in the United States, between 20% and 58% of youth and young adults reported favourable beliefs about menthol’s sensory effects, particularly refreshment sensations, which were associated with increased intention to use and current use of menthol cigarettes [12]. Taken together, the literature suggests that smokers may develop misperceptions about the risks of smoking menthol cigarettes due to the distinct and favourable physiological and sensory effects of menthol [16].

Much research has been undertaken in the United States where the prevalence of menthol use is high amongst current smokers (39% among those aged 12 and over in 2012–2014), especially among non-Hispanic Black smokers (85%) [19]. There has been relatively little research about menthol smokers’ sensory experiences and associated beliefs in nations with a lower prevalence of menthol use. Banning menthol in such countries would result in small but important public health benefits and could also help to provide the precedent that may support countries with higher rates of menthol use to enact similar bans. Such bans are consistent with World Health Organization recommendations that nations prohibit menthol and its analogues, precursors or derivatives in cigarettes and other tobacco products in order to prevent youth uptake of smoking and facilitate cessation [20,21].

Australia is one such country that has a relatively low prevalence of menthol use and may therefore be well placed to follow the lead of countries including Turkey [22], Ethiopia [23], Canada [24], Moldova [22] and the United Kingdom and the European Union [25], which have all recently implemented a ban on menthol in tobacco products, and the United States, where the Food and Drug Administration has announced it is working towards issuing a proposed product standard to ban menthol as a characterising flavour in cigarettes and cigars [26]. In a national study conducted in Australia between 2012 and 2014, the prevalence of menthol use among adult tailor-made cigarette smokers (tailor-made cigarettes can also be referred to as factory-made or manufactured cigarettes) was just under 12% [27]. More recently, a 2016 population telephone survey in the Australian state of Victoria indicated that 15% of adult current smokers sometimes or always smoked menthol cigarettes [28]. In addition to these low rates of menthol use, several aspects of Australia’s comprehensive tobacco control program may mean Australian menthol smokers’ sensory experiences and associated beliefs differ from those experienced or held by smokers in other jurisdictions. Australian tobacco control measures include mass media campaigns, smoke-free legislation, graphic health warnings, access to cessation aids and a series of tax increases over the past 11 years. In 2006, Australia was among the first countries to ban the cigarette brand variant descriptors ‘light’ and ‘mild’, after the Australian Competition and Consumer Commission deemed these descriptors had misled or deceived some smokers into viewing cigarettes marketed in this way as less harmful than other cigarette variants [29]. Australia was the first country in the world to implement tobacco plain packaging in 2012, a measure that severely limited tobacco companies’ ability to use packaging as a marketing tool [30] but which also led to a rapid increase in the new product innovations introduced to the Australian market [31].

These tobacco control measures—individually and collectively—could influence smokers’ product choices. The sensory experiences and related beliefs reported by Australian menthol smokers may therefore differ from those observed in other jurisdictions. For this reason, we examined Australian smokers’ sensory experiences of menthol and non-menthol cigarettes and their beliefs about how damaging and how enjoyable they find cigarettes offering these different sensory experiences.

## 2. Materials and Methods

### 2.1. Participants

The study sample comprised 999 Australians aged 18 to 69 years who were at least weekly smokers of tailor-made cigarettes, roll-your-own tobacco or both types of products. Respondents were recruited in July 2019 through a non-probability online panel accredited under the International Organization for Standardization standards for Market Research (ISO 2052). Panel members opt in to receive invitations to participate in research and receive points from the panel managers for each survey they complete; points may be accumulated and redeemed for gift vouchers and other rewards.

Quotas were applied to achieve a sex and age distribution consistent with the sample of at least weekly smokers in the 2016 Australian National Drug Strategy Household Survey [32]: 25.5% 18- to 29-year-olds, 45.0% 30- to 49-year-olds, and 29.5% 50- to 69-year-olds. Within each age group, there were equal quotas for males and females within a 10% tolerance.

Panel members received an email invitation to participate with a survey weblink. Potential participants were screened based on sex, age, smoking frequency and product use to fulfil eligibility criteria and the above quotas.

### 2.2. Menthol Product Use

Participants were first asked to indicate if they had ever smoked “*tailor-made cigarettes (sometimes called factory-made or manufactured cigarettes)*” and “*roll-your-own cigarettes*”. Ever users were then asked how often they currently smoked tailor-made and/or roll-your-own cigarettes: “*daily*”, “*at least weekly*”, “*less than weekly*”, “*not at all*” or “*don’t know/can’t say*”. Only those participants who selected “*daily”* or “*at least weekly*” for tailor-made and/or roll-your-own cigarettes continued with the survey.

In two separate questions, eligible respondents were then asked to select the statement that best described their use of menthol tailor-made cigarettes and menthol roll-your-own cigarettes: “*I have never tried them*”, “*I have tried them, but have never smoked them regularly*”, “*I used to smoke them regularly, but do not smoke them now*”, “*I currently smoke them regularly”* or “*don’t know/can’t say*”, where “regularly” was defined as using them at least once a week. Participants who had at least tried menthol tailor-made and/or menthol roll-your-own cigarettes were then asked further questions about their menthol-related sensory experiences. These same participants were also asked questions about the sensory experiences associated with smoking non-menthol cigarettes, irrespective of whether they had ever tried them.

### 2.3. Sensory Experiences

Participants rated the sensory experiences of smoking menthol and non-menthol cigarettes, respectively, on a 100-point Visual Analogue Scale (VAS). These sensory experiences were drawn from previous research [1,8]. Each VAS was anchored with polar adjectives to measure each sensory experience: “*Smooth—Harsh*”, “*Soothing on the throat—Irritating on the throat*”, *“Fresh taste—Tobacco taste*” and “*Clean—Dirty*” (see Figure 1). A marker was positioned at the midpoint of 50, and participants were instructed to move the marker in either direction. Participants had to move the marker off the midpoint but could return it to 50 to indicate that their experience of the two sensations was equal or neutral; we used a 5-point margin around the midpoint to indicate responses that did not reflect a substantially stronger experience of one sensation or the other. Responses were dichotomised for analyses, with the category of interest being ratings between 1 and 44 (i.e., “More smooth than harsh”, “More soothing than irritating”, etc.). Scores between 45 and 100 and *“don’t know, can’t say”* responses were coded as the opposite category (i.e., “Neutral/More harsh than smooth”, “Neutral/More irritating than soothing”, etc.).

### 2.4. Beliefs about Sensory Experiences

Participants were asked about their perceptions of the damage caused by cigarettes with the four different sensory experiences described above. All questions followed the same phrasing and response format, for example: “*How damaging do **smooth** cigarettes feel compared to harsh cigarettes?*”, with the response options comprising (i) “***Smooth***
*cigarettes feel a lot more damaging*”, (ii) *“**Smooth** cigarettes feel a bit more damaging*”, (iii) “***Smooth***
*and harsh cigarettes feel equally damaging*”, (iv) “***Harsh***
*cigarettes feel a bit more damaging*”, (v) “***Harsh***
*cigarettes feel a lot more damaging*” and (vi) “*Don’t know/can’t say*”. Responses were then collapsed into four categories: the first category *“**smooth** more damaging than harsh”* comprised the responses (i) “***smooth***
*cigarettes feel a lot more damaging*” and (ii) *“**smooth** cigarettes feel a bit more damaging*”; the second category comprised the response (iii) “***smooth***
*and **harsh** cigarettes feel equally damaging*”; the third category “***harsh***
*more damaging than smooth*” comprised the responses (iv) “***harsh***
*cigarettes feel a bit more damaging*” and (v) “***harsh***
*cigarettes feel a lot more damaging*”; and the fourth category comprised the (vi) “*don’t know/can’t say*” responses (see Figure 2).

For the present series of analyses, our primary interest was in responses where the negative sensory experience was perceived as more damaging (e.g., *“**harsh** more damaging than **smooth***”), which we interpreted as indicating that the positive sensory experience was perceived as less damaging (e.g., if ***harsh***
*is more damaging than **smooth,*** then ***smooth***
*is less damaging than **harsh***). Using this approach, we were able to indirectly assess whether smokers hold misperceptions about the harmfulness of cigarettes that produce that sensory experience, without requiring them to directly state that some types of cigarettes are *less* damaging than others. We used the terms “feel” and “damaging” in these questions to obtain a more sensory or experiential measure of smokers’ perceptions of different cigarettes’ relative harmfulness, rather than a more cognitive measure such as “*How harmful are smooth cigarettes compared to harsh cigarettes?”*. Preliminary work conducted for this project—which included exploratory focus groups with 121 smokers and cognitive testing of questionnaire items with a convenience sample of 18 smokers—revealed that these smokers: (i) frequently used the term “damage” when discussing the relative harmfulness of different tobacco products; and (ii) understood the term “damage” to cover both short-term and long-term harms. However, we also note that the word “damage” may have greater resonance for Australian smokers than for smokers in other countries, given its use in the landmark National Tobacco Campaign that aired in Australia between 1997 and 2001 and prominently featured the tagline “Every Cigarette is Doing You Damage” [33].

Participants were also asked about their enjoyment of cigarettes with respect to the four sensory experiences. The question format was like that described above, for example: “*How enjoyable are **smooth** cigarettes compared to **harsh** cigarettes?*”, with the response options comprising (i) “ ***Smooth***
*cigarettes are a lot more enjoyable*”, (ii) *“**Smooth** cigarettes are a bit more enjoyable*”, (iii) “***Smooth***
*and **harsh** cigarettes are equally enjoyable*”, (iv) “***Harsh***
*cigarettes are a bit more enjoyable*”, (v) “***Harsh***
*cigarettes are a lot more enjoyable*” and (vi) “*Don’t know/can’t say*”. Responses were then collapsed into four categories for the analyses: the first category *“smooth more enjoyable than harsh”* comprised the responses (i) “***smooth***
*cigarettes are a lot more enjoyable*” and (ii) *“**smooth** cigarettes are a bit more enjoyable*”; the second category comprised the response (iii) “***smooth***
*and **harsh** cigarettes are equally enjoyable”*; the third category “*harsh more enjoyable than smooth*” comprised the responses (iv) “***harsh***
*cigarettes are bit more enjoyable*” and (v) “***harsh***
*cigarettes are a lot more enjoyable*”; the fourth category comprised the (vi) “*don’t know/can’t say”* response. For the present analyses, our primary interest was in the “***smooth***
*more enjoyable than **harsh***” category.

### 2.5. Statistical Analyses

Data were analysed using *Stata* version 16.1 (StataCorp, 2019). Aside from our description of the sample, the present analyses focused on current menthol smokers only (*n* = 252), defined as those who answered “*I currently smoke them regularly”* to at least one of the two questions asking them to describe their use of menthol tailor-made/roll-your-own cigarettes. However, of the 252 current menthol smokers, 7 (2.8%) responded with “*don’t know/can’t say”* for all four sensory experience questions, leaving 245 current menthol smokers in the analytic sample.

We used odds ratios from McNemar’s test to determine whether smokers were more likely to associate each positive sensory experience with menthol cigarettes, compared to their likelihood of associating the positive sensory experience with non-menthol cigarettes. *p*-values adjusted for multiple testing within each sensory experience were calculated using the Holm method [34] to control the family-wise error rate to 5%. This approach performs well in controlling the family-wise error rate irrespective of the degree of dependence between outcomes or the effect size [35].

## 3. Results

In total, 999 smokers were surveyed, and 758 (75.9%) were identified as having ever smoked menthol cigarettes (i.e., ‘ever’ users). There was a significant association between age and ever menthol cigarette use (*p* = 0.021). Smokers aged 18 to 29 years were 1.73 times more likely to be ever menthol smokers (80.6%) compared to those aged 50 to 69 years (70.6%; OR = 1.73, 95% CI = 1.16, 2.58, *p* = 0.007) but were not significantly more likely to be ever menthol smokers compared to those aged 30 to 49 years (76.7%; OR = 1.26, 95% CI = 0.87, 1.85, *p* = 0.223).

Amongst all 999 smokers surveyed, current use of menthol cigarettes was 25.2% (*n* = 252). There was also a significant association between age and current menthol cigarette use (*p* < 0.001). Smokers aged 18 to 29 years were 3.50 times more likely to be current menthol smokers (34.0%) compared to those aged 50 to 69 years (12.8%; OR = 3.50, 95% CI = 2.70, 5.37, *p* = < 0.001). Likewise, smokers aged 30 to 49 years were 2.70 times more likely to be current menthol smokers (28.4%) than those aged 50 to 69 years (OR = 2.70, 95% CI = 1.81, 4.02, *p* = < 0.001).

### 3.1. Demographic Characteristics

Table 1 shows the demographic characteristics of current menthol smokers in the sample: the gender distribution was balanced, and the largest proportion of participants was aged 30 to 49 years (50.8%, compared with 34.1% of 18- to 29-year-olds and 15.1% of 50- to 69-year-olds; *M* age = 36.8, *SD* = 11.98). Most current menthol smokers (84.9%) had some tertiary education.

### 3.2. Sensory Experiences

Figure 3 shows the ratings for each sensory experience, separately for menthol and non-menthol cigarettes. For each sensory experience, a greater proportion of current menthol smokers associated the favourable sensory experience with menthol cigarettes than with non-menthol cigarettes. Relative to non-menthol cigarettes, a greater proportion rated menthol cigarettes as more smooth than harsh (19.2% for non-menthol vs. 46.5% for menthol cigarettes), more soothing than irritating on the throat (18.0% vs. 43.3%), more fresh than tobacco tasting (53.9% vs. 80.0%) and more clean than dirty tasting (21.6% vs. 46.1%).

Consistently, odds ratios from McNemar’s Test (Table 2) indicated that current menthol smokers were 5.79 times more likely to experience menthol cigarettes as more smooth than harsh than they were to experience non-menthol cigarettes as more smooth than harsh. They were also 8.75 times more likely to experience menthol cigarettes as more soothing than irritating on the throat than they were to experience non-menthol cigarettes as more soothing than irritating on the throat. They were 4.37 times more likely to experience menthol cigarettes as tasting fresh rather than tasting like tobacco and 5.29 times more likely to experience menthol cigarettes as feeling clean rather than feeling dirty.

### 3.3. Beliefs about Sensory Experiences

Overall, as shown in Figure 4, around one-third of current menthol smokers reported the “correct” belief that there is no difference in how damaging smooth and harsh cigarettes, soothing and irritating cigarettes, fresh-tasting and tobacco-tasting cigarettes and clean-feeling and dirty-feeling cigarettes are. Fewer smokers perceived these different sensory experiences as being equally enjoyable; rather, the more positive sensory experience tended to be associated with greater enjoyment.

Amongst current menthol smokers, 29.8% believed that harsh cigarettes are *more* damaging than smooth cigarettes, which we interpret as indicating that these smokers believe smooth cigarettes are less damaging than harsh cigarettes. About three-quarters (75.9%) believed that smooth cigarettes are *more* enjoyable than harsh cigarettes.

A similar pattern was seen for beliefs about cigarettes that feel soothing versus irritating on the throat. Over a quarter (29.0%) of current menthol smokers believed that cigarettes which feel irritating on the throat are *more* damaging than cigarettes that feel soothing on the throat, which can also be interpreted as showing that these smokers believe that cigarettes that feel soothing are less damaging. More than two-thirds (69.4%) also believed that cigarettes that feel soothing on the throat are *more* enjoyable.

Similarly, 26.9% of current menthol smokers believed that cigarettes with a tobacco taste are *more* damaging than cigarettes with a fresh taste, which we interpret as showing that these smokers believe that cigarettes with a fresh taste are less damaging. Almost three-quarters (71.8%) also believed that cigarettes with a fresh taste are more enjoyable.

Finally, 22.4% of current menthol smokers believed that cigarettes that feel dirty are *more* damaging than cigarettes that feel clean, which we interpret as showing that these smokers believe that clean-feeling cigarettes are less damaging. Over two-thirds (69.8%) of current menthol smokers also believed that clean-feeling cigarettes are *more* enjoyable.

## 4. Discussion

In this sample of Australian at least weekly smokers, three-quarters had at least some experience of smoking menthol cigarettes, and a quarter (25%) currently smoked menthol cigarettes at least once a week. Ever use and current use of menthol cigarettes in our sample was most common among young adults (18- to 29-year-olds). This is consistent with findings from the United States [19,36] and with the evidence that menthol in cigarettes is associated with increased initiation and progression to regular cigarette smoking [1,8]. This is potentially because the sensory properties of menthol increase young people’s interest in trying cigarettes and mean that, when they do try, they find them easier to smoke [11,12]. Current menthol smokers were more likely to rate menthol cigarettes as providing positive sensory experiences than non-menthol cigarettes. Menthol smokers were about four to nine times more likely to experience menthol than non-menthol cigarettes as smooth, soothing on the throat, fresh-tasting and clean-feeling. Between 22% (clean feeling) and 30% (smooth) incorrectly perceived cigarettes with these favourable sensations as being less damaging than cigarettes with the opposite more aversive sensory experiences of harshness, irritating on the throat, tobacco-tasting and dirty-feeling. In addition, between 69% (soothing on the throat) and 76% (smooth) perceived the positive sensory experiences to be more enjoyable.

The pattern of results is consistent with research from New Zealand [11] and the United States [9,12], where menthol smokers also report experiencing menthol cigarettes as smoother, more soothing on the throat, fresher tasting and cleaner feeling than non-menthol cigarettes. Furthermore, in Malaysia and Thailand [18], between 16% and 36% of smokers, respectively, agreed that menthol cigarettes were less harmful based on the perception of smoother smoke. The positive sensory experiences that smokers associate with menthol cigarettes may be due to menthol’s cooling, anaesthetic and analgesic properties that can improve the palatability of tobacco [6,17] and mask the harshness of the smoke [15]. The present findings also suggest that the physiological sensory effects of menthol may be causing smokers to believe that tobacco products with smooth, soothing, fresh-tasting and clean-feeling properties are less damaging and more enjoyable. The enjoyment associated with these positive sensory experiences may be in part attributed to the continued pairing of the specific sensory experience with nicotine intake during smoking, which may lead to nicotine dependence [37] and can reduce successful smoking cessation [8].

The results of the present study should be interpreted in the context of some limitations. Firstly, the survey used an online opt-in non-probability sample, although quotas were applied to achieve a representative sex and age distribution consistent with the 2016 Australian National Drug Strategy Household Survey [32] sample of at least weekly smokers. We do not claim that the prevalence estimate for smokers of menthol cigarettes or the findings related to sensory experiences and beliefs about damage and enjoyment are representative of the Australian population of menthol smokers. In particular, the proportion of current menthol smokers in our sample who had completed at least some tertiary education (85%) was substantially higher than that observed in a national sample of daily smokers (of all types of cigarettes; around 50% had completed any tertiary education [38]). This likely reflects the younger age of menthol smokers. It is also important to note that the question measuring menthol use did not specify whether this included cigarettes with menthol capsules or “crushballs” in the filter. Such cigarettes have rapidly grown in popularity since they were first introduced into the Australian market [39,40]: between 2012 and 2014, the preference among Australian adult smokers for cigarettes with flavour capsules increased significantly from 1% to 3% [41], and in 2014 more than half of Australian adolescent past-month smokers reported having smoked a menthol capsule cigarette [42]. Given the strong global growth of this cigarette variety [40,43,44,45], use among Australian adult smokers is likely to be substantially higher now than found in 2014. It is possible that smokers interpreted our questions about menthol cigarettes as also applying to menthol capsule cigarettes, which could partly account for the higher rates of current menthol use observed in this sample (25%) compared with previous surveys (12% in 2012–2014 [27] and 15% in 2016 [28]). This increase in the use of menthol capsule cigarettes may also explain why our sample of current menthol smokers comprised a similar proportion of males and females. Previous studies have found menthol cigarette use more common among females than males [19,28]—most likely due to historical efforts from the tobacco industry to actively target females as menthol cigarette consumers [3]. However, more recent evidence generally suggests that capsule cigarettes may be similarly appealing to females and males [41,42], with some exceptions [44].

Another limitation is that we did not measure current menthol smokers’ experience smoking non-menthol cigarettes. As a result, their ratings of the sensory experiences of smoking non-menthol cigarettes will reflect a combination of expectations and varying experiences. Similarly, we probed smokers’ beliefs about damage and enjoyment for cigarettes with different sensory experiences in general and not specifically for menthol cigarettes (or non-menthol cigarettes). Some smokers’ beliefs about how damaging and/or enjoyable certain sensory experiences may thus be influenced by their experience of smoking non-menthol cigarettes, to the extent that they also experience these as smooth, soothing, fresh-tasting and clean-feeling.

The above limitations notwithstanding, the present findings can inform new public health interventions focused on reframing the misperception—held by approximately one-third of smokers in the current study—that cigarettes with positive sensory experiences are potentially less harmful. Potential interventions could include new warning labels for tobacco packaging and multi-media campaigns to reframe the positive sensations smokers experience when consuming menthol cigarettes, such as those previously used in California and other jurisdictions [46,47,48]. For example, package warnings could more directly confront smokers by explaining that the sensory effects of menthol mask their ability to sense potential smoking-related damage or how menthol’s pharmacological effects in the brain increase nicotine dependence [37], which in turn may hinder their efforts to quit [8].

From a broader policy perspective, the present findings support the World Health Organization’s call to ban menthol based on several studies suggesting that the removal of menthol from cigarettes would have significant public health benefits [21]. Recent international studies demonstrate that enacting bans on menthol has reduced menthol cigarette sales [49] and total cigarette sales [50] and increased the likelihood of smoking cessation [51].

## 5. Conclusions

In summary, the present study suggests that the sensory effects of menthol may contribute to the development of misperceptions about the reduced risk of menthol cigarettes, while also contributing to increased enjoyment of smoking. The development of tobacco package warning labels and/or multi-media campaigns that reframe these sensory experiences may reduce misperceptions of the potential for reduced risk and undermine the enjoyment that smokers experience when smoking menthol cigarettes, helping these smokers move closer to quitting. In addition, a ban on the use of menthol in tobacco products in Australia would eliminate the source of these sensory experiences and, as has occurred elsewhere [51,52], could be expected to increase quitting behaviours.

## Figures and Tables

**Figure 1 ijerph-18-05501-f001:**
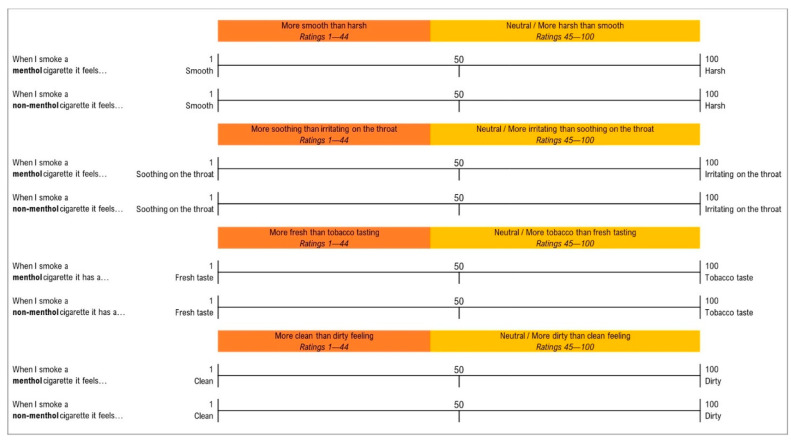
Sensory experience rating scales.

**Figure 2 ijerph-18-05501-f002:**
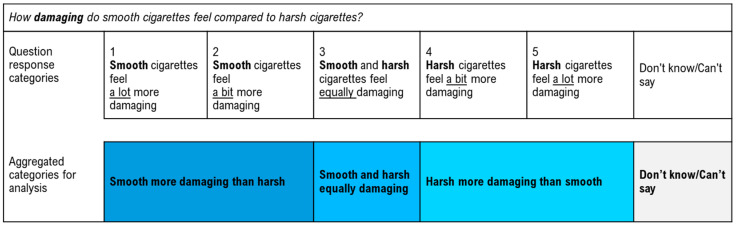
Example of the rating scale for beliefs about damage for the smooth to harsh sensory experience.

**Figure 3 ijerph-18-05501-f003:**
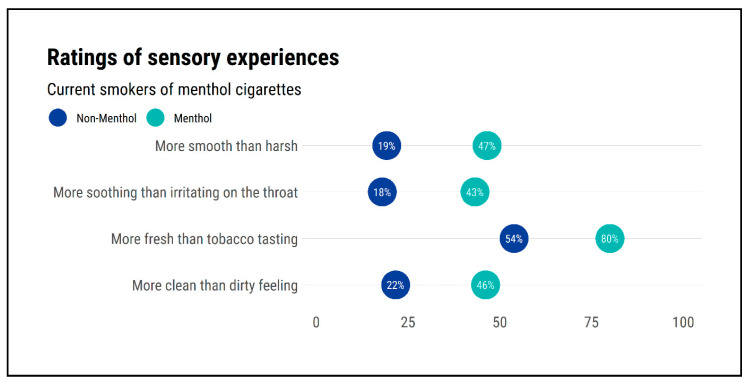
Proportion of current menthol smokers who associated each favourable sensory experience with menthol cigarettes and non-menthol cigarettes (*N* = 245).

**Figure 4 ijerph-18-05501-f004:**
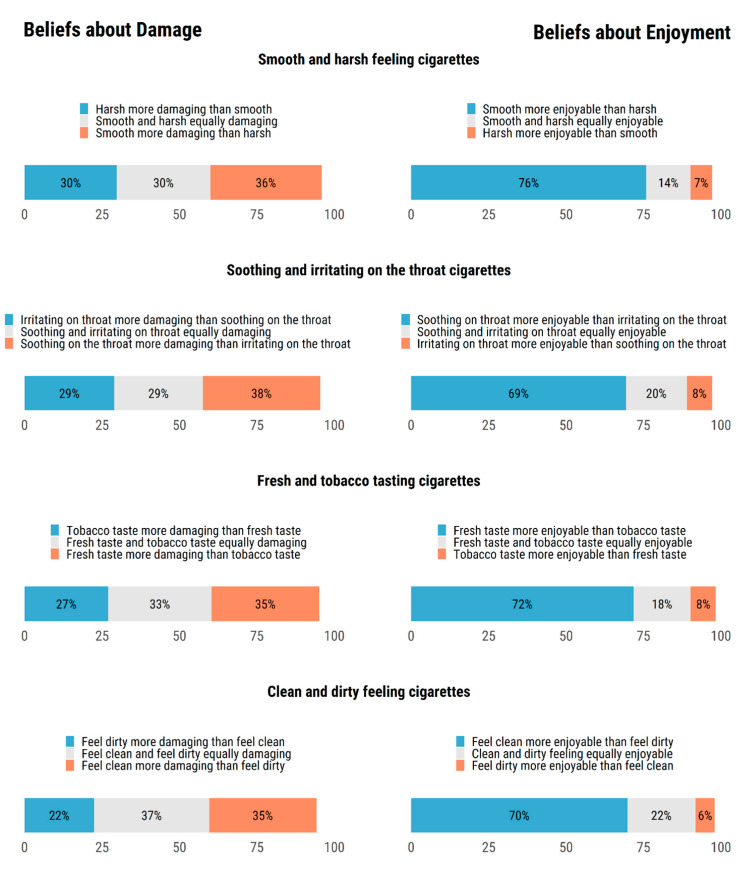
Beliefs about damage and enjoyment related to each sensory experience amongst current menthol smokers (*N* = 245). Note. Data for “don’t know/can’t say” response categories are not displayed.

**Table 1 ijerph-18-05501-t001:** Demographic characteristics of current menthol smokers (*N* = 252).

Demographic Characteristic	*n* (%)
Gender ^a^	
Male	132 (52.4)
Female	120 (47.6)
Age	
18 to 29	86 (34.1)
30 to 49	128 (50.8)
50 to 69	38 (15.1)
Highest level of education	
Some tertiary education	214 (84.9)
No tertiary education	36 (14.3)
Prefer not to say	2 (0.8)

^a^ Participants were asked: “Which gender do you identify with?” and were able to respond: male; female; other; prefer not to say. No participants selected “other” or “prefer not to say”.

**Table 2 ijerph-18-05501-t002:** Cross-tabulation frequencies, cell percentages, and test statistics for sensory experiences of menthol and non-menthol cigarettes for current menthol smokers.

	Non-Menthol Cigarettes		McNemar’s Test			
**Menthol Cigarettes**	More smooth than harsh	More harsh than smooth	OR	[95% CI]	*p*	*p* _adjusted_
More smooth than harsh	33 (13.5%)	81 (33.1%)	5.79	[3.26, 11.05]	<0.001	<0.001
More harsh than smooth	14 (5.7%)	117 (47.8%)				
	More soothing than irritating on the throat	More irritating than soothing on the throat	OR	[95% CI]	*p*	*p* _adjusted_
More soothing than irritating on the throat	36 (14.7%)	70 (28.6%)	8.75	[4.20, 21.06]	<0.001	<0.001
More irritating than soothing on the throat	8 (3.3%)	131 (53.5%)				
	More fresh than tobacco tasting	More tobacco than fresh tasting	OR	[95% CI]	*p*	*p* _adjusted_
More fresh than tobacco tasting	113 (46.1%)	83 (33.9%)	4.37	[2.63, 7.62]	<0.001	<0.001
More tobacco than fresh tasting	19 (7.8%)	30 (12.2%)				
	More clean than dirty feeling	More dirty than clean feeling	OR	[95% CI]	*p*	*p* _adjusted_
More clean than dirty feeling	39 (15.9%)	74 (30.2%)	5.29	[2.96, 10.14]	<0.001	<0.001
More dirty than clean feeling	14 (5.7%)	118 (48.2%)				

Note. *N* = 245, OR = odds ratio, and *p*_adjusted_ = Holm adjusted *p*-value for multiple testing (*k* = 4).

## Data Availability

The data presented in this study may be available upon reasonable request to the corresponding author.

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
