# Peer review of "Australian Smokers’ Sensory Experiences and Beliefs Associated with Menthol and Non-Menthol Cigarettes"

_ijerph, 2021, doi:10.3390/ijerph18115501_

Round 1

Reviewer 1 Report

The authors have addressed previous concerns either with thorough revisions or with appropriate explanation and justification of the limitations.  I have no further concerns about the present form of the manuscript.

Reviewer 2 Report

Thankyou for the opportunity to review your resubmitted manuscript. I congratulate your team for their work!

This manuscript is a resubmission of an earlier submission. The following is a list of the peer review reports and author responses from that submission.

Round 1

Reviewer 1 Report

This paper seeks to extend the now substantial body of evidence on consumers’ sensory and risk perceptions of menthol cigarettes to an Australian population. The authors use the potential for the introduction of a regulatory ban on mentholated cigarettes in Australia as the context for the current work. The findings are consistent with prior research showing that smokers of menthol cigarettes perceive them as smoother and with lower health risks compared with non-mentholated products, despite no difference in the actual health risk.

Overall, the paper is well written, with a strong methodological approach and appropriate analysis. The conclusions are generally appropriately justified. However, two substantial concerns should be addressed.

  1. The final sample was selected from an initial sample of N=999 adult smokers, of whom 76% reported having previously used a menthol product. The final sample included just N=252 regular (weekly) smokers of menthol cigarettes. The major study outcomes focused on these participants’ ratings of sensory experiences of mentholated cigarettes vs. non-mentholated cigarettes. This is potentially problematic because it is not known to what extent the sample has had experience using non-mentholated cigarettes, making the question largely hypothetical and prone to response bias. Presumably, regular menthol smokers persist with mentholated cigarettes because they are perceived as smooth and less risky. The results, then, are hardly surprising. However, a more informative comparison might have involved comparison of the subjective ratings of regular menthol smokers vs. infrequent/past menthol smokers. A substantial proportion of smokers in the original sample have previously used a menthol product but did not continue to smoke menthol cigarettes regularly: do these smokers also view menthol cigarettes as smoother and less risky, or are these perceptions attenuated in this group? The findings of this alternative analysis may generate deeper insight into menthol sensory perceptions and product preference, and may not be as clear cut as the current analysis suggests. The conclusion that similar proportions of participants perceive menthol as less harmful as more harmful (and neutral) may not be particularly strong evidence in support of a menthol ban.

  1. While the reason for the study – to generate evidence in support of a regulatory ban on menthol cigarettes in Australia – is sound, the rationale for the specific research question is rather poorly developed. The authors point to consistent, robust evidence showing that menthol smokers perceive their products as smoother, misleading them to perceive them as less risky. Australia is a viable regulatory target because of the low prevalence of menthol use. However, there is no clear rationale for the need to re-confirm the perceptions of menthol cigarettes in an Australian sample. What unique evidence is required to support a ban in Australia, and how will these data inform regulators about the likely impact of a menthol ban in Australia?

Minor issues

There appears to be some confusion with the use of terms to describe the health impact of smoking. While harm is a perfectly acceptable term, it is more often used to describe the population impact. Health risk (or risk) might be more appropriate in some places (e.g. line 63). The issue becomes more of a concern with the introduction of the term damaging, which seems rather imprecise. Was this term explained to participants - i.e. how did they understand this term? Did participants distinguish between short term (e.g. throat irritation) and longer term disease outcomes?

Line 78: “precedence”: should be precedent(?)

Line 80: Explain “tailor-made” at first use (explained helpfully at line 111, but might lessen confusion for an international audience if this occurs earlier)

The reason for the 5-point margin used at the mid-point of the VAS scale should be explained, ideally with a supporting citation

Figure 1: As is, it contains little informative detail. Consider revising to capture the 4 sensory experiences. This is important, because the 4 experiences are referred to later but Fig 1 misleadingly describes only menthol vs. non-menthol ratings.

Line 341: suggest adding the word “public”, i.e. “would have significant public health benefits”. The examples provide in the following sentence speak to population level impact rather than specific health outcomes.

The final sentence of the conclusion appears unsupported by the present evidence: how do these data support the conclusion that a menthol ban will improve health outcomes? Which health outcomes?  Will smoking rates actually decline? How do the present data inform this conclusion?

Author Response

REVIEWER 1

This paper seeks to extend the now substantial body of evidence on consumers’ sensory and risk perceptions of menthol cigarettes to an Australian population. The authors use the potential for the introduction of a regulatory ban on mentholated cigarettes in Australia as the context for the current work. The findings are consistent with prior research showing that smokers of menthol cigarettes perceive them as smoother and with lower health risks compared with non-mentholated products, despite no difference in the actual health risk.

Overall, the paper is well written, with a strong methodological approach and appropriate analysis. The conclusions are generally appropriately justified. However, two substantial concerns should be addressed.

The final sample was selected from an initial sample of N=999 adult smokers, of whom 76% reported having previously used a menthol product. The final sample included just N=252 regular (weekly) smokers of menthol cigarettes. The major study outcomes focused on these participants’ ratings of sensory experiences of mentholated cigarettes vs. non-mentholated cigarettes. This is potentially problematic because it is not known to what extent the sample has had experience using non-mentholated cigarettes, making the question largely hypothetical and prone to response bias. Presumably, regular menthol smokers persist with mentholated cigarettes because they are perceived as smooth and less risky. The results, then, are hardly surprising. However, a more informative comparison might have involved comparison of the subjective ratings of regular menthol smokers vs. infrequent/past menthol smokers. A substantial proportion of smokers in the original sample have previously used a menthol product but did not continue to smoke menthol cigarettes regularly: do these smokers also view menthol cigarettes as smoother and less risky, or are these perceptions attenuated in this group? The findings of this alternative analysis may generate deeper insight into menthol sensory perceptions and product preference, and may not be as clear cut as the current analysis suggests. The conclusion that similar proportions of participants perceive menthol as less harmful as more harmful (and neutral) may not be particularly strong evidence in support of a menthol ban.

ResponseThank you for the opportunity to respond to this thought-provoking suggestion. If our research question aimed to compare perceptions of menthol and non-menthol cigarettes among users of both variants, we would agree that the analyses Reviewer 1 has suggested would be appropriate. However, our research question aimed to inform regulation regarding the use of menthol in cigarettes and to guide the development of communication interventions that could correct smokers’ misperceptions about menthol cigarettes (see (now) Lines 400 – 407), interventions which are both intended to impact those currently smoking menthol cigarettes and those who may do so in the future, not those who have previously tried menthols but found this variant was not to their liking. The rationale for a ban and development of communication interventions needs to be informed by data on how current menthol smokers experience these cigarettes, and whether the unique sensory experiences associated with smoking menthol cigarettes contribute to risk perceptions and smoking enjoyment. 

We agree with Reviewer 1’s comment that regular menthol smokers persist with mentholated cigarettes because they perceive these as smooth (alongside other favourable sensory experiences measured in this study) and therefore less risky. This association is exactly why these perceptions are of concern and why regulators in several jurisdictions are considering policies to restrict menthol use. As explained in response to Comment #2 (Reviewer 1) and Comment #14 (Reviewer 2), we have revised our manuscript to provide additional information explaining the need to measure perceptions of menthol cigarettes among Australian smokers in 2019, even though research from other jurisdictions exists.

We acknowledge that we did not measure current menthol smokers’ experiences of smoking non-menthol cigarettes. This is a limitation of the study and we agree that we cannot determine how much expectations and/or experiences influenced respondents’ ratings of their sensory experiences with non-menthol cigarettes. Nonetheless, these ratings still enable us to assess whether current menthol smokers experience these cigarettes more favourably than they have experienced (or expect to experience) non-menthol cigarettes. We thus do not believe this limitation undermines our contribution; nonetheless, we acknowledge this limitation at Lines 388 – 391:

“Another limitation is that we did not measure current menthol smokers’ experience smoking non-menthol cigarettes. As a result, their ratings of the sensory experiences of smoking non-menthol cigarettes will reflect a combination of expectations and varying experiences.”

2. While the reason for the study – to generate evidence in support of a regulatory ban on menthol cigarettes in Australia – is sound, the rationale for the specific research question is rather poorly developed. The authors point to consistent, robust evidence showing that menthol smokers perceive their products as smoother, misleading them to perceive them as less risky. Australia is a viable regulatory target because of the low prevalence of menthol use. However, there is no clear rationale for the need to re-confirm the perceptions of menthol cigarettes in an Australian sample. What unique evidence is required to support a ban in Australia, and how will these data inform regulators about the likely impact of a menthol ban in Australia?

Response: In response to this comment, and Comment #14 (Reviewer 2), we have reworked and expanded the introduction (see changes at (now) Lines 82 – 120) to highlight several aspects of Australia’s comprehensive tobacco control program. Attributes of this program may mean that Australian menthol smokers have different reasons for choosing the products they smoke and we acknowledge that the sensory experiences they associate with menthol and non-menthol cigarettes may differ from those observed in other jurisdictions. The section that justifies the need for this study (the final two paragraphs of the introduction) now reads:

“Australia is one such country that has a relatively low prevalence of menthol use and may therefore be well placed to follow the lead of countries including Turkey [1], Ethiopia [2], Canada [3], Moldova [1], and the United Kingdom and the European Union [4], which have all recently implemented a ban on menthol in tobacco products. In a national study conducted in Australia between 2012 and 2014, the prevalence of menthol use among adult tailor-made cigarette smokers (tailor-made cigarettes can also be referred to as factory-made or manufactured cigarettes) was just under 12% [5]. More recently, a 2016 population telephone survey in the Australian state of Victoria indicated that 15% of adult current smokers sometimes or always smoked menthol cigarettes [6]. In addition to these low rates of menthol use, several aspects of Australia’s comprehensive tobacco control program may mean Australian menthol smokers’ sensory experiences and associated beliefs differ from those experienced or held by smokers in other jurisdictions. Australian tobacco control measures include mass media campaigns, smokefree legislation, graphic health warnings, access to cessation aids, and a series of tax increases over the past 11 years. In 2006, Australia was among the first countries to ban the cigarette brand variant descriptors ‘light’ and ‘mild’, after the Australian Competition and Consumer Commission deemed these descriptors had misled or deceived some smokers into viewing cigarettes marketed in this way as less harmful than other cigarette variants [7] Australia was the first country in the world to implement tobacco plain packaging in 2012, a measure that severely limited tobacco companies’ ability to use packaging as a marketing tool [8]; but which also led to a rapid increase in the new product innovations introduced to the Australian market [9].

These tobacco control measures—individually, and collectively—could influence smokers’ product choices. The sensory experiences and related beliefs reported by Australian menthol smokers may therefore differ from those observed in other jurisdictions. For this reason, we examined Australian smokers’ sensory experiences of menthol and non-menthol cigarettes, and their beliefs about how damaging and how enjoyable they find cigarettes offering these different sensory experiences.”

Minor issues

3. There appears to be some confusion with the use of terms to describe the health impact of smoking. While harm is a perfectly acceptable term, it is more often used to describe the population impact. Health risk (or risk) might be more appropriate in some places (e.g. line 63). The issue becomes more of a concern with the introduction of the term damaging, which seems rather imprecise. Was this term explained to participants - i.e. how did they understand this term? Did participants distinguish between short term (e.g. throat irritation) and longer term disease outcomes?

Response: We have replaced the term “harm” with “risk” in several places. See changes at Lines 29, 30 (“lower risk of harm” changed to “lower level of risk”), 60, 65, 72, 416, and 420.

In the manuscript, we have explained why we used the terms “feel” and “damaging” in questions measuring perceptions of different tobacco products’ relative harmfulness of. At Lines 201 – 212, we have added the following text:

“We used the terms “feel” and “damaging” in these questions to obtain a more sensory or experiential measure of smokers’ perceptions of different cigarettes‘ relative harmfulness.  rather than a more cognitive measure such as “How harmful are smooth cigarettes compared to harsh cigarettes?” would not have encouraged consideration of sensory experiences. Preliminary work conducted for this project—which included exploratory focus groups with 121 smokers and cognitive testing of questionnaire items with a convenience sample of 18 smokers—revealed that these smokers: (i) frequently used the term “damage” when discussing the relative harmfulness of different tobacco products; and (ii) understood the term “damage” to cover both short-term and long-term harms. However, we also note that the word “damage” may have greater resonance for Australian smokers than for smokers in other countries, given its use in the landmark National Tobacco Campaign that aired in Australia between 1997 and 2001 and prominently featured the tagline “Every Cigarette is Doing You Damage” [32].”

4. Line 78: “precedence”: should be precedent(?)

Response: Thank you for picking up this typo. “precedence” has been changed to “precedent” on (now) Line 81.

5. Line 80: Explain “tailor-made” at first use (explained helpfully at line 111 but might lessen confusion for an international audience if this occurs earlier).

Response: As suggested, at (now) Lines 98-99 we have added an explanation that “(tailor-made cigarettes can also be referred to as factory-made or manufactured cigarettes)”. We have retained the explanation provided at (now) Line 141, given it illustrates how tailor-made cigarettes were defined to study participants.

6. The reason for the 5-point margin used at the mid-point of the VAS scale should be explained, ideally with a supporting citation.

Response: The 5-point margin around the mid-point of the VAS scale was used to capture those responses that did not indicate a substantially stronger experience of one sensation or the other; e.g., if on a scale of 1 “smooth” to 100 “harsh”, a participant recorded a response of 45, 46, 47, 48 or 49, we did not feel confident categorising that respondent as experiencing their cigarettes as “more smooth than harsh”. We have revised the sensory experiences sub-section of the Materials and Methods section to explain our rationale for this approach. In addition to a few minor edits at (now) Lines 160 and 168, we have added the following sentence at (now) Lines 162 – 167:

“A marker was positioned at the midpoint of 50, and participants were instructed to move the marker in either direction. Participants had to move the marker off the midpoint, but could return it to 50 to indicate that their experience of the two sensations was equal or neutral; we used a 5-point margin around the midpoint to indicate responses that did not reflect a substantially stronger experience of one sensation or the other.”

7. Figure 1: As is, it contains little informative detail. Consider revising to capture the 4 sensory experiences. This is important, because the 4 experiences are referred to later but Fig 1 misleadingly describes only menthol vs. non-menthol ratings.

Response: As suggested, we have revised Figure 1 to illustrate how the four sensory experiences were measured. We believe this change will help readers understand how we measured these four variables. Due to the increased size of this figure, we have positioned it on its own page, rotated to preserve the landscape orientation of the figure. We have also revised the figure title accordingly.

8. Line 341: suggest adding the word “public”, i.e. “would have significant public health benefits”. The examples provide in the following sentence speak to population level impact rather than specific health outcomes.

Response: Thank you for this suggestion. The word “public” has been added to (now) Line 410.

9. The final sentence of the conclusion appears unsupported by the present evidence: how do these data support the conclusion that a menthol ban will improve health outcomes? Which health outcomes? Will smoking rates actually decline? How do the present data inform this conclusion?

Response: We have softened the final sentence of the conclusion so that it now reads (at Lines 421 – 424):

“In addition, a ban on the use of menthol in tobacco products in Australia would eliminate the source of these sensory experiences and, as has occurred elsewhere [ref], could be expected to increase quitting behaviours.”

Reviewer 2 Report

Thankyou for the opportunity to review your manuscript. I particularly enjoyed the ways you presented your results. I have a few suggestions:

Abstract:

Do you need to repeat the point about 'smoother, more soothing on throat, fresher tasting and cleaner feel'? Could you say 'pleasant sensory effects on lines 17-18 and then use the full phrase on lines 25-26?

Introduction:

remove 'are' from line 40.

Is 'perception' a better word than 'association' (which has different connotation) on line 52?

Similarly, is 'perceived association' or 'perceived link' better than 'association' line 57?

Villanti et al state that menthol use among participants of non-Hispanic Black ethnicity was high (84.6%, rather than non-Hispanis and Black.

The introduction would benefit from clearly stating the novelty of your research.

Is there a gender difference among menthol smokers that needs to be mentioned?

Method:

Can you please state the amount of financial incentive?

Table 1:

Could 'not male' be separated into female and other/non-binary?

Results:

Please consider removing the sub-sentences related to your interpretation of the findings (lines 251-269) and place them in the discussion. Then in the discussion you could talk about what these findings mean in relation to awareness of harm and public health campaigns.

LImitations:

Please consider adding a point about the educational attainment among your sample and compare to average educational attainment of smokers at a population level. What might this mean to your findings?

Conclusion:

Does 'lead to the development' need to be changed to 'may contribute to' or 'be associated with' to avoid the suggestion of cause and effect?

Author Response

Thankyou for the opportunity to review your manuscript. I particularly enjoyed the ways you presented your results. I have a few suggestions.

10. Abstract: Do you need to repeat the point about 'smoother, more soothing on throat, fresher tasting and cleaner feel'? Could you say 'pleasant sensory effects on lines 17-18 and then use the full phrase on lines 25-26?

Response: Thank you – that’s a great suggestion. We have changed this text at Lines 17 – 19 so that it now reads “…which can lead smokers to have favourable sensory experiences.” As suggested, we have retained the full phrase at Line 26.

11. Introduction: remove 'are' from line 40.

Response: “are” has been removed from the sentence in (now) Line 41.

12. Introduction: Is 'perception' a better word than 'association' (which has different connotation) on line 52? Similarly, is 'perceived association' or 'perceived link' better than 'association' line 57?

Response: We have revised this section of the manuscript to ensure our meaning is as clear as possible. The sentence beginning at (now) Line 53 now reads:

“The effects of this historical marketing persist as many menthol smokers continue to incorrectly believe that menthol cigarettes are safer or less harmful [10-14].”

The first sentence of the next paragraph, beginning at (now) Line 58 now reads:

“In addition to the effects of marketing, these misperceptions may also arise from the impact that taste and other sensory experiences have on smokers’ perceptions of risk [15].”

13. Introduction: Villanti et al state that menthol use among participants of non-Hispanic Black ethnicity was high (84.6%), rather than non-Hispanic and Black.

Response: Thank you for picking up this typo. At (now) Line 77 the “and” has been removed so that this now reads “non-Hispanic Black smokers”.

14. Introduction: The introduction would benefit from clearly stating the novelty of your research.

Response: In response to this comment, and Comment #2 (Reviewer 1), we have expanded the introduction (Lines 102 – 114) to highlight Australia’s comprehensive tobacco control program. Several of these tobacco control measures may mean Australian menthol smokers have different reasons for choosing the products they smoke such that the sensory experiences they associate with menthol and non-menthol cigarettes may differ from those observed in other jurisdictions. The section that justifies the need for this study (the final two paragraphs of the introduction) is included in our response to Comment #2 (Reviewer 1).

15. Introduction: Is there a gender difference among menthol smokers that needs to be mentioned?

Response: In the third paragraph of the Discussion (Lines 381 – 387), we have added a reference to historical gender differences in menthol cigarette use and have noted how the rapid growth of menthol “crushball” cigarettes may have contributed to the higher rate of current menthol use in this sample than in previous Australian studies. We note that our sample of current menthol smokers included a similar proportion of males and females, contrary to what might have been expected based on historical patterns. However, recent studies with adolescents (White et al., 2015) and adults (Thrasher et al., 2016) have indicated that capsule cigarettes may be similarly appealing to males and females. This added text reads:

“This increase in the use of menthol capsule cigarettes may also explain why our sample of current menthol smokers comprised a similar proportion of males and females. Previous studies have found menthol cigarette use more common among females than males [19,23]—most likely due to historical efforts from the tobacco industry to actively target females as menthol cigarette consumers [3]. However, more recent evidence generally suggests that capsule cigarettes may be similarly appealing to females and males [39,40], with some exceptions [42].”

16. Method: Can you please state the amount of financial incentive?

Response: We do not know the amount of financial incentive offered to survey participants, as the online panel provider manages panel recruitment and compensation. The reimbursements offered are determined by several factors, including the length and complexity of the survey and the anticipated difficulty in meeting recruitment quotas. However, we know that reward is provided to participants in the form of points that accumulate and can later be redeemed for gift vouchers and other rewards. We now explain this process in the Participants sub-section of the Materials and Methods section. The sentence: “This panel consists of individuals who opt-in and receive small financial incentives for survey participation” has been replaced with this sentence at (now) Lines 128-130:

“Panel members opt-in to receive invitations to participate in research and receive points from the panel managers for each survey they complete; points may be accumulated and redeemed for gift vouchers and other rewards.”

17. Table 1: Could 'not male' be separated into female and other/non-binary?

Response: The “Not male” label has been changed to “Female” in Table 1, as upon rechecking the data we discovered that none of the participants in this study selected the “other” or “prefer not to say” response options in the gender question. We have added a footnote to Table 1 to explain this:

“Participants were asked: “Which gender do you identify with?” and were able to respond: male; female; other; prefer not to say. No participants selected “other” or “prefer not to say”.”

18. Results: Please consider removing the sub-sentences related to your interpretation of the findings (lines 251-269) and place them in the discussion. Then in the discussion you could talk about what these findings mean in relation to awareness of harm and public health campaigns.

Response: We appreciate that some language is unusual for a results section; however, we believe these comments are needed to assist correct interpretation of the results. As explained at (now) Lines 198 – 201, we wanted to avoid asking smokers to state that cigarettes with certain sensory experiences were less damaging than other types of cigarettes. Our preliminary focus group discussions with smokers suggested that it is easier for them to acknowledge that they perceive cigarettes with unfavourable sensory experiences as more harmful or damaging, than it is to report that cigarettes with favourable sensory experiences are less harmful or damaging. We suspect this preference may be because they know that the “correct answer” is that “all cigarettes are harmful”, influenced perhaps by exposure to Australia’s National Tobacco Campaign which featured the tagline “Every Cigarette is Doing You Damage” (such that reporting that some cigarettes might not be as harmful feels “incorrect”).

To measure whether smokers held the misperception that (for example) smooth cigarettes are less damaging than harsh cigarettes, we therefore asked them to indicate whether Harsh cigarettes feel a bit/lot more damaging”. If a smoker experiences harsh cigarettes as feeling more damaging than smooth cigarettes, then we inferred that they also experienced smooth cigarettes as feeling less damaging than harsh cigarettes. Given our interest lies in smokers’ inferences (smooth are less damaging than harsh), we believe it is important to retain the original structure of the results section. 

19. Limitations: Please consider adding a point about the educational attainment among your sample and compare to average educational attainment of smokers at a population level. What might this mean to your findings?

Response: We are not aware of any nationally representative data on the educational attainment of adult menthol smokers in Australia, and so have compared the educational attainment among our sample of regular menthol smokers (85% with at least some tertiary education) to that observed in a national sample of daily smokers of all types of cigarettes (around 50% in the 2017-18 National Health Survey). The higher level of tertiary educational attainment in our sample likely reflects the younger age of menthol smokers, although without comparison to a nationally representative sample of menthol smokers we cannot be sure if it instead reflects an oversampling of tertiary educated respondents in our opt-in non-probability sample. It is therefore difficult to speculate about any implications of this educational distribution for interpretation of our findings.

We have added two sentences to the limitations section to acknowledge that the proportion of current menthol smokers in our sample who had completed at least some tertiary education is substantially higher than that observed in a national sample of smokers of all types of cigarettes. At (now) Lines 365 – 369, this addition reads:

“In particular, the proportion of current menthol smokers in our sample who had completed at least some tertiary education (85%) was substantially higher than that observed in a national sample of daily smokers (of all types of cigarettes; around 50% had completed any tertiary education [36]). This likely reflects the younger age of menthol smokers.”

20. Conclusion: Does 'lead to the development' need to be changed to 'may contribute to' or 'be associated with' to avoid the suggestion of cause and effect?

Response: Thank you for this helpful suggestion. We have changed “may lead to the development of” at (now) Line 416 to “may contribute to…”. We have also revised the end of this sentence to clarify that it is the sensory effects rather than perceptions of reduced harm that may contribute to increased enjoyment of smoking. The final part of this sentence has changed from “which may contribute to increased enjoyment of smoking” to “while also contributing to increased enjoyment of smoking”. Of course, it is possible that perceptions of reduced harm contribute to smokers’ enjoyment of smoking, but our findings do not speak directly to this association.